# Utilization of digital tools to enhance COVID-19 and tuberculosis testing and linkage to care: A cross-sectional evaluation study among *Bodaboda* motorbike riders in the Nairobi Metropolis, Kenya

**Simon Karanja**[1][*], **Jane Aduda**[1], Reuben Thuo[1], Fred Wamunyokoli[1], Philip Oyier[1], Gideon Kikuvi[1], Henry Kissinger[1], John Gachohi[1], Patrick Mburugu[1], David Kamau[1], Joseph Matheri[1], Susan Mwelu[1], Joseph Machua[2], Patrick Amoth[3], Denver Mariga[3], Ian Were[3], Musa Mohamed[4], Judith Kimuyu[5], Samson Saigilu[6], Rose Wangeci[7], Kevin Mubadi[8], Joseph Ndung'u[9], Khairunisa Suleiman[9], Rigveda Kadam[9], Paula Akugizibwe[9]

1 Jomo Kenyatta University of Agriculture and Technology, Juja, Kenya, 2 Kenya Forestry Research Institute, Nairobi, Kenya, 3 Ministry of Health Kenya, Nairobi, Kenya, 4 Department of Health Services, Nairobi Metropolitan Services, Nairobi, Kenya, 5 Department of Health Services Machakos County, Machakos, Kenya, 6 Department of Health Services Kajiado County, Nairobi, Kenya, 7 Department of Health Services Kiambu County, Kiambu, Kenya, 8 Bodaboda Safety Association of Kenya, Nairobi, Kenya, 9 FIND, Geneva, Switzerland

☯ These authors contributed equally to this work.
* skaranja@jkuat.ac.ke

## Abstract

Kenya has registered over 300,000 cases of COVID-19 and is a high-burden tuberculosis country. Tuberculosis diagnosis was significantly disrupted by the pandemic. Access to timely diagnosis, which is key to effective management of tuberculosis and COVID-19, can be expanded and made more efficient through integrated screening. Decentralized testing at community level further increases access, especially for underserved populations, and requires robust systems for data and process management. This study delivered integrated COVID-19 and tuberculosis testing to commercial motorbike (Bodaboda) riders, a population at increased risk of both diseases with limited access to services, in four counties: Nairobi, Kiambu, Machakos and Kajiado. Testing sheds were established where riders congregate, with demand creation carried out by the Bodaboda association. Integrated symptom screening for tuberculosis and COVID-19 was conducted through a digital questionnaire which automatically flagged participants who should be tested for either, or both, diseases. Rapid antigen-detecting tests (Ag-RDTs) for COVID-19 were conducted onsite, while sputum samples were collected and transported to laboratories for tuberculosis diagnosis. End-to-end patient data were captured using digital tools. 5663 participants enrolled in the study, 4946 of whom were tested for COVID-19. Ag-RDT positivity rate was 1% but fluctuated widely across counties in line with broader regional trends. Among a subset tested by PCR, positivity was greater in individuals flagged as high risk by the digital tool (8% compared with 4% overall). Of 355 participants tested for tuberculosis, 7 were positive,

**Data Availability Statement:** All relevant data are within the manuscript and its Supporting Information files.

**Funding:** The study was funded by FIND, United Kingdom (FCDO 40105983), Switzerland (81066910), Netherlands (SDD 4000004160), Canada (DFATD 7429348), The Kingdom of Saudi Arabia (FIND – ACT-A DX PARTNERSHIP 20.08.2020), The Rockefeller Foundation (2020 HTH 059), Germany (BMZ Covid-19 Diagnostic and Surveillance Response 27.07.2021), Australia (DFAT 76442), Kuwait (M239/2020), The Government of Portugal and Partners (ANF, BCP, CGF, APIFARMA) and The BlackRock Foundation (Grant Agreement as of April 20, 2022). Medical writing support was funded by FIND, according to Good Publication Practice guidelines. Beyond the authors listed in this paper, the funders had no role in study design, data collection and analysis, decision to publish, or preparation of the manuscript.

**Competing interests:** J.N., K.S., R.K., and P.A. declare that they are employed by FIND. The other authors have declared that no competing interests exist.

with the resulting prevalence rate higher than the national average. Over 40% of riders had elevated blood pressure or abnormal sugar levels. The digital tool successfully captured complete end-to-end data for 95% of all participants. This study revealed high rates of undetected disease among Bodaboda riders and demonstrated that integrated diagnosis can be delivered effectively in communities, with the support of digital tools, to maximize access.

## Introduction

The COVID-19 pandemic, caused by severe acute respiratory syndrome coronavirus 2 (SARS-CoV-2), has had an unprecedented impact on the health, economic and social well-being of Kenyans [1, 2]. As of February 2023, Kenya had reported over 340,000 COVID-19 cases and 5600 deaths from the disease [3]. The diversion of resources to the pandemic response and restrictions on movement have also created challenges for the management of other diseases, including the long-standing epidemics of tuberculosis (TB) and HIV in the country [4–7].

Kenya has one of the highest national TB prevalence rates, and TB-HIV coinfection rates, in the world. The World Health Organization (WHO) estimates that there were 133,000 incident cases of TB in Kenya in 2021 [8], and 33,000 deaths [8]. Half of the estimated incident cases went undetected or unreported, contributing to a high case fatality ratio, which is currently estimated at 1 in 4 [8].

Kenya's 2019–2023 National Strategic Plan for Tuberculosis, Leprosy and Lung Health emphasises finding missing cases of TB, with the target of diagnosing and treating 597,000 people by 2023 [9]. This is key to sustaining decreases in case fatality rates, which declined by 44% between 2015 and 2020. However, the Ministry of Health's 2022 End Term Review of the NSP (2019–2023) has highlighted that disruptions attributable to COVID-19 threaten this progress. A study on retrospective data from health facilities in Nairobi between 2019 and 2021 found a 31% reduction in TB diagnosis after the pandemic began and demonstrated the effectiveness of programmatic interventions in improving TB case detection and treatment outcomes [4].

Integration of TB and COVID-19 management has been recommended to mitigate the disruptive impact of the pandemic response on TB services, while increasing efficiencies. This is particularly true for diagnosis, given the overlap in clinical presentation of TB and COVID-19, which presents opportunities to increase testing for both diseases through simultaneous screening [10, 11]. Simultaneous screening of COVID-19 and TB can be delivered through community-based testing models, supported by digital tools for integrated screening and data management, to optimize use of resources and access to testing [12].

Equitable access to effective diagnostic testing and a rapid turnaround time of test results, to enable individuals to self-manage or be linked to care as needed, are key to controlling COVID-19 [13]. However, Kenya's existing capacities for diagnostic testing have been outpaced by the demand for testing during the pandemic. With an estimated population of 53.8 million people, the country had conducted approximately 3.8 million tests by the time of conclusion of this study [14, 15]–representing an average daily COVID-19 testing rate in Kenya of 0.1 tests per 1000 people, which is 10 times lower than WHO testing targets of 1 test per 1000 people [16]. Limited testing has resulted in the substantial underreporting of cases and deaths in the country [17, 18].

As of early 2022, most of the COVID-19 tests conducted in Kenya had been reverse transcriptase polymerase chain reaction (RT-PCR) tests, with limited use of rapid antigen-

detecting tests (Ag-RDTs). Although RT-PCR tests are the gold standard for detecting SARS-CoV-2, they are slower (typical turnaround time of 24 to 48 hours), expensive, and require robust infrastructure, an elaborate supply chain to support testing services and well-trained laboratory staff [13, 19]. In contrast, Ag-RDTs are faster (providing results usually in under 30 minutes), easier to use, more affordable, and can be conducted at the point of care without laboratory infrastructure [13, 19]. These features mean that Ag-RDTs have the potential to dramatically increase access to decentralized COVID-19 testing in the community.

However, expanding testing outside of health facilities also presents specific challenges, including the management and transmission of data between the testing site and the central health system, and between the site and the client. The latter is particularly challenging when dealing with mobile populations who may not reside near the testing site but need to be followed up if positive to ensure appropriate linkage to care and contact tracing.

Digital tools can help to address these challenges, by establishing reliable patient databases with end-to-end data, automating transmission of data between different sections of the health system and patients, and remote monitoring or follow-up of patients. Digital tools can also offer clinical decision-making support to triage clients for testing. As a result, there has been considerable interest in using digital tools to inform decision-making and support the management of people with infectious diseases such as COVID-19 and TB [20–25], particularly when facilitating greater access in the community.

Certain populations in Kenya are at particularly high risk of contracting COVID-19 and TB in the community. Studies suggest that presence or history of TB increases the risk of SARS-CoV-2 infection, while TB/SARS-CoV-2 co-infection is associated with more rapid and severe symptom development and disease progression with poor outcomes for both diseases [26]. The national TB survey conducted in Kenya in 2016 indicated that the highest burden of TB is seen in the urban slums (with 760 cases per 100,000 people) and among people aged 25–34 years (with 716 cases per 100,000).

Bodaboda taxi riders, who transport passengers on motorbikes from one location to another, are typically low-wage men aged 22–45 years, who lack medical insurance and have limited access to healthcare. The majority of Bodaboda riders within the Nairobi Metropolis also reside in informal settlements with poor hygiene and sanitation facilities and high rates of TB and are thus among the populations with a higher risk of TB. As their occupation requires interacting with large numbers of people daily, they are also at particularly high risk of contracting and transmitting COVID-19 and TB in the community.

Kenya's Ministry of Health (MoH) has identified Bodaboda riders as a population of interest, due to their increased risk for COVID-19 and the fact that they have not received the same degree of targeted interventions for COVID-19 screening, testing and vaccination as workers in other sectors such as health, education, security, and long-distance truck drivers. It is therefore imperative to create awareness and demand among the Bodaboda riders for COVID-19 and TB testing and include this critical group in healthcare planning and targeted service delivery.

This study aimed to demonstrate and evaluate the use of digitally supported community-based testing models to deliver integrated diagnostic services for COVID-19, TB and non-communicable diseases to this underserved population in the Nairobi Metropolis. In this study, an existing digital tool that had been deployed for COVID-19 community symptom screening and referral was adapted for use in screening, data capture, and transmission of test results. The study also measured disease rates in the Bodaboda population, assessed performance of Ag-RDTs in this setting, and generated operational insights that could strengthen future community-based testing models.

## Methods

### Study design and population

This was a cross-sectional evaluation study to demonstrate the use of digital platforms in enhancing COVID-19 and TB testing and linkage to care among Bodaboda riders in the Nairobi Metropolis. The Nairobi metropolitan area includes Nairobi County and bordering centres from Kiambu, Machakos and Kajiado Counties. The study area has a total population of around 9.34 million people, with 125,000 Bodaboda riders registered in Nairobi County and over 100,000 riders across the other three counties.

Eligible participants were registered Bodaboda riders aged ≥18 years, operating within the Nairobi Metropolis, who presented themselves at testing sites. All participants had to provide written consent. Bodaboda riders previously registered in the study and who had been on TB treatment in the last 24 months were excluded from the study. Participants (Bodaboda riders) were recruited between 21 October 2021 and 1 February 2022 and their samples were collected between the same dates. The authors did not have access to information that could identify individual participants during data collection.

Follow-up of participants was undertaken through county-patient management mechanisms. For participants who tested positive for COVID-19 via PCR, follow-up was undertaken by Sub-County Disease Surveillance Coordinators who managed result communication and linkage to care; participants were followed up for 10–14 days. For TB positive cases, follow-up was carried out through the Sub-County TB Coordinators for purposes of enrolment into care, tracing for individuals in contact with TB cases, and screening and testing of those in contact with the TB case at the household level.

### Study objectives

The primary objective was to evaluate the use of digital tools in conjunction with Ag-RDTs to support decentralized COVID-19 and TB screening, testing, contact tracing and linkage to care of Bodaboda riders in the Nairobi metropolis.

Other secondary objectives were also investigated in the study, including:

1. creating awareness and demand for COVID-19 and TB testing among Bodaboda riders using rider-led mobilization and digital messaging;

2. determining the COVID-19 and TB positivity and co-infection rates among the Bodaboda riders;

3. evaluating the accuracy/reliability of the Ag-RDT compared with RT-PCR.

### Digital health tools

The study employed various digital solutions to support the testing initiatives. The primary tool was an adaptation of the Kenya COVID-19 Tracker app–an existing tool developed by Medic Mobile using the Community Health Toolkit Core Framework (CHT Core Framework) and deployed by Kenya's MoH. CHT Core is an open-source application that is highly configurable to meet emerging programme needs. For the purposes of this study, the tool was adapted by Jomo Kenyatta University of Agriculture and Technology (JKUAT) with support from software developers Medic Mobile (Kenya hub) and the CHT peer community, Nairobi Kenya.

This tool was selected following a mapping of digital solutions deployed in the COVID-19 response in Kenya, which was conducted by JKUAT and FIND, the global alliance for diagnostics, in 2021. In order to avoid introducing new tools to an already fragmented digital

landscape, the study sought to identify existing digital solutions that could be leveraged to support the key functionalities needed. CHT was selected for several reasons: it included most of these functions and could be modified to meet additional requirements and it had already been used in communities in Kenya for COVID-19 screening and other areas for disease management. CHT is also endorsed by the MoH, is open-source, and offers an online community of practice for support with troubleshooting.

CHT includes a backend dashboard for data collation and analysis, and a mobile app on the front end for data capture and clinical decision-making support. The dashboard was activated for this study and used to generate aggregates to allow remote monitoring of study progress. Data were collected from the front end and synchronized to the server every evening.

The existing contact registration form and case investigation form were customized from the Kenya COVID-19 Tracker app, and screening questions for TB and COVID-19 (S1 Table) were added for the purposes of the study. Adaptations included addition of fields at the steps for registration, symptom screening and testing. As blood sugar, blood pressure and oxygen measurements were also taken from all participants, fields to capture these data were further integrated in the tool. After pilot testing and subsequent modifications, the app was rolled out for use across the study sites. Data were collected in real-time and synchronized a minimum of once a day, with indicators for the overall project and each of the four counties monitored through the dashboard. To encourage complete data, validation checks built into the digital tool required study personnel to complete all fields before proceeding with or exiting the workflow, with the exception of COVID-19 Ag-RDT results which could be noted as pending and updated after the waiting time of 15 minutes for the Ag-RDTs.

Several other digital tools/platforms were used in this study for end-to-end data capture and transmission:

1. Registration of participants and screening for eligibility for TB or COVID-19 testing was conducted on the adapted Kenya COVID-19 Tracker app (described previously). The app also provided clinical decision-making support to determine whether individuals should be tested for COVID-19, TB, or both.

2. Data from the Ag-RDTs were entered into the Kenya COVID-19 Tracker app for onward transmission into MoH databases.

3. All testing data were uploaded on the county LabWare laboratory information management system (LIMS) to be fed into the national daily tallies.

4. All TB cases were managed on TIBU and outcomes at 2 months of testing were transmitted back to the study investigators. TIBU is an Android application designed to digitize lung health reporting and routine surveillance, currently used in Kenya to digitize TB data.

## Sampling sites

Mapping of Bodaboda sheds, where riders congregate, and of health facilities was conducted in each county. The location of community-based sampling sites for the study was determined based on proximity to both the sheds and the health facilities. For each sampling site, the study team constituted the following persons: a Clinical Officer, a nurse, two laboratory technicians, two data clerks, Bodaboda association personnel to mobilize riders and JKUAT project staff.

## Demand creation

Bodaboda riders were encouraged to participate in the study through campaigns led by officials of the Bodaboda Safety Association of Kenya (BAK). Messages were jointly crafted by the

study team and BAK officials, and were disseminated digitally (via bulk SMSs and WhatsApp) and through posters displayed in spaces frequented by the target population. Field officers working with the study and BAK leaders also engaged directly with Bodaboda riders to disseminate messaging. Some of the campaign messages shared information that:

i.  Free medical check-ups (blood pressure, blood sugar, body mass index [BMI]) were available for all study participants.

ii.  Linkage to further treatment services, where needed, would be provided for symptomatic cases of COVID-19 by county Clinical Officers.

iii.  Sensitization sessions would be conducted by National Social Security Fund (NSSF) and National Hospital Insurance Fund (NHIF) to encourage enrollment, and NHIF registration with three months' subscription would be offered to all positive COVID-19 cases.

In addition, sensitization flyers were circulated via WhatsApp and physically through the BAK leadership at the pick-up points. A total of 7680 messages were sent out by SMS and WhatsApp. Field officers were also sent to the respective pick-up points to share information about the study with Bodaboda riders.

## Study process

An overview of the study procedures is shown in Fig 1 and described further in this section.

The study procedures began with the consenting participants completing an informed consent form (ICF) after orientation by the study team. Bodaboda sheds were randomized while the riders presented themselves for screening and testing. After providing consent, participants were given a free general medical check-up which involved measurement of random blood sugar levels, blood pressure, temperature, height and weight measurements. This served to identify potential cases of hypertension and diabetes, in accordance with national guidance on screening and linkage to care [27]. It also helped to generate demand for testing, led by BAK, by providing a broader package of services that Bodaboda riders would be interested in. Following the general medical check-up, participants who met the clinical parameters for further care for non-communicable diseases like diabetes were subsequently provided with advice and referred to care where indicated.

Participants were then screened for eligibility to undergo symptom screening and testing for COVID-19, TB or both. There was strict adherence to the eligibility criteria. Anyone who was currently on treatment for TB was excluded from TB testing. Those who were eligible were then moved to the next step, while those not eligible were released and advised on COVID-19 and TB infection prevention and control measures. The next step involved bidirectional screening for both COVID-19 and TB, using the bidirectional algorithm which was built into the Kenya COVID-19 Tracker app. This integrated a series of questions on symptoms or risk factors common to both COVID-19 and TB, including additional questions specific to each disease, in line with MoH guidelines. The screening questions included as part of the case investigation form are shown in S1 Table.

Based on their responses, each individual was classified as a suspected COVID-19 case, presumptive TB case, both a suspected COVID-19 and presumptive TB case, or having none of the symptoms that were being investigated. Participants with none of the symptoms were also released and advised on COVID-19 and TB infection prevention and control measures.

Sputum samples were collected from those who were identified as presumptive TB cases and samples were sent to the participating health facility laboratories for GeneXpert analysis. All presumptive TB cases were also tested for COVID-19. Nasopharyngeal samples were

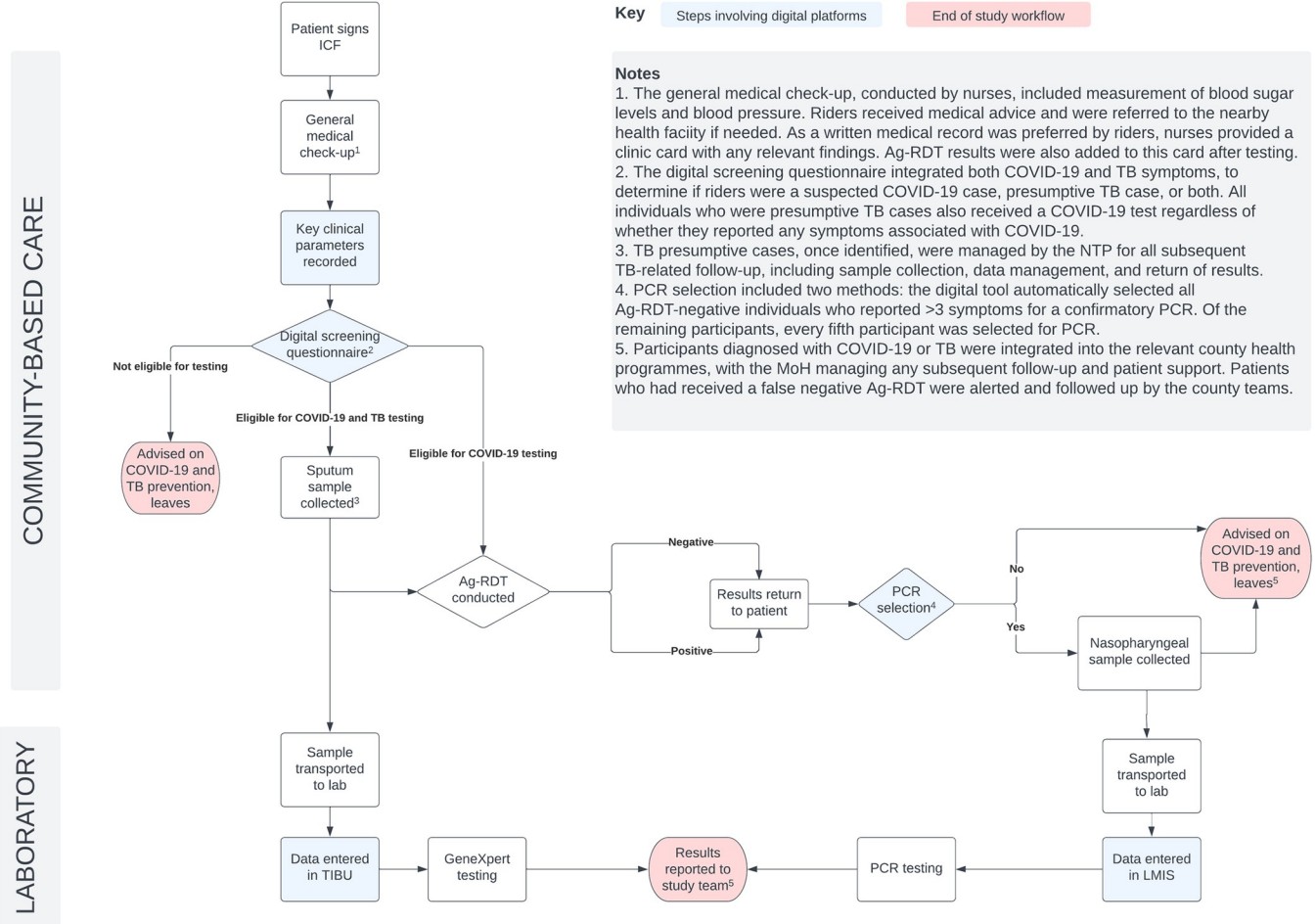

**Fig 1. Algorithm for study.** Ag-RDT, rapid antigen-detecting tests; ICF, informed consent form; LMIS, laboratory information management system; PCR, polymerase chain reaction; TB, tuberculosis.

collected from all those who were eligible for COVID-19 testing. All the samples were tested using an Ag-RDT (PANBIO™ COVID-19 Ag RAPID test device), conducted on-site by nurses according to manufacturer's instructions. Of participants undergoing COVID-19 testing via the Ag-RDTs, 20% were also randomly selected for confirmatory testing using RT-PCR. In addition, samples from participants deemed to have a high-risk profile for COVID-19 (those who were positive for four or more symptoms of COVID-19), were automatically selected by the app for testing with RT-PCR.

Sampling followed infection prevention and control requirements as recommended by the MoH, with strict adherence to the manufacturer's standard operating procedures/instructions for use. Specimens for both COVID-19 and TB were stored in cool boxes with ice packs until they were analysed or transported to various laboratories for analysis. Nasopharyngeal swabs for COVID-19 RT-PCR and sputum samples for GeneXpert analysis were transported in cool boxes to the respective country hospital laboratories using designated riders from the MoH.

Those who tested negative for TB and COVID-19 on GeneXpert, COVID-19 Ag-RDT and PCR, where conducted, were released and advised on COVID-19 and TB preventive measures, while those who tested positive were linked to care through the county health structures.

## Reporting and follow up

All samples collected for testing were tracked using a unique sample identifier generated by the Kenya COVID-19 Tracker app to facilitate onward processing. This unique identification was stored in the database and used by the different digital platforms to exchange information through application programming interfaces. All COVID-19 Ag-RDT results were captured on the Kenya COVID-19 Tracker app and transmitted to County Health Management Teams for onward reporting to the national databases. The results from the RT-PCR tests were also relayed to the study team through the laboratory personnel from participating health facilities.

Where RT-PCR results were discordant with the Ag-RDT results–particularly when a sample tested positive under RT-PCR but negative under the Ag-RDT–the results were conveyed to the respective County Disease Surveillance Coordinator who was responsible for tracing the clients and linking them to care when necessary.

In addition, such results were transmitted to the Public Health Emergency Operation Centre (PHEOC) to be included in the daily national tallies. Participants placed on home-based isolation were required to self-register on the Jitenge system, although the study team was unable to access the database to directly monitor outcomes of this group. Results from TB tests conducted at the county hospital laboratories were communicated back to the study team through the respective County-TB Coordinators.

## Sample size and analysis

The study area under consideration is made up of 32 sub-counties with approximately 231,500 Bodaboda riders. A sample of Bodaboda riders from all these sub-counties was enrolled in the study. The study aimed to conduct 160 Ag-RDT COVID-19 tests per sub-country, for a total of 5120 tests. The study also aimed to collect 1260 samples for RT-PCR testing.

Data were collected using the digital tool with pre-coded variables. Various metrics were computed on the number of participants at each stage of the study, including the number of participants who visited the sites and underwent screening for COVID-19 and TB, and the proportion of participants with complete records captured in the digital tool. Also recorded was the medical background of participants such as of blood sugar and blood pressure levels. Results from the COVID-19 and TB tests were recorded in terms of the number testing positive, negative and inconclusive. COVID-19 positivity rates were recorded for Ag-RDT and RT-PCR testing methods, while TB positivity rates were recorded based on GeneXpert analysis of sputum samples. Ag-RDT test performance was calculated in terms of sensitivity, specificity, positive predictive value and negative predictive value, evaluated against the "gold standard" RT-PCR. Symptoms were also recorded among those testing positive for COVID-19 and TB.

Microsoft Excel, R and Stata were used to analyse data. Confounding was addressed mainly at the study design stage. In particular, as the study focused on a specific group (male Bodaboda riders mainly between 25 to 45 years of age in the Nairobi Metropolis), sex, occupation, age and geographical location were largely controlled. Nasal sample collection was conducted by laboratory officers/technologists who had received training on sample collection from trainers (government laboratory officers who had previously been trained on COVID-19 testing), in line with national guidelines. PCR confirmatory tests were carried out in centralized, national accredited government diagnostic laboratories in designated level 4 and 5 health facilities, which ensured standardized testing. Further, data collection was carried out by personnel who were trained using standard protocols and guidelines, supervised by trainers.

### Ethical considerations

The study received ethical clearance and approval from the JKUAT-Institutional Ethical Review committee (JKUAT-IREC). Participants had the right to withdraw from the study at any time. The study was conducted according to ICH GCP E6/R2 guidelines [28] and the Declaration of Helsinki [29]. To ensure the confidentiality and anonymity of study subjects, data were extracted and de-identified and stored in specific password-protected computers.

## Results

A total of 5663 participants visited the sites and underwent a medical check-up and bidirectional screening for COVD-19 and TB. Of those who underwent screening, 4946 (87.3%) met the criteria for testing for COVID-19 only, or both COVID-19 and TB. Table 1 shows the demographics of study participants. The average age of participants was 33.7 years, with 61% of participants aged between 25 and 44 years. Nearly all participants in the study were male (97.1%, n = 5498); there were 165 female participants. Most of the participants were from Nairobi County, with the smallest number of participants from Kajiado County.

### Proportion of participants with complete records captured in the digital tool

Of the 4946 participants eligible for COVID-19 testing, one participant declined to take the test after consenting, citing fears of discomfort from the nasopharyngeal swab procedure. Therefore, 4945 participants were tested for COVID-19 and 95% (4699) had testing results captured in the system. Of those tested, 246 test results were missing due to data entry errors, as data clerks forgot to update results in the app. These missing test results were excluded from the final data analysis. In the <5% of cases where the checks did not manage to prevent capture of erroneous data, these fields were identified by study personnel and erroneous data were corrected. All TB samples had results recorded in TIBU and reported back to the study team through the respective county TB coordinators.

### Medical background

A total of 51.6% of the participants had normal blood sugar levels (4–7.8 mmol/L), 5.17% of the participants had raised blood sugars (more than 7.8 mmol/L) and 43.2% had below-normal

**Table 1. Demographic characteristics of study participants.**

| Gender/County | Age (years) | | | | | | |
|---|---|---|---|---|---|---|---|
| | Below 24 | 25-34 | 35-44 | 45-54 | 55+ | Missing | Total |
| **Female** | 26 | 59 | 39 | 23 | 6 | 12 | 165 |
| *Kajiado* | | 3 | 1 | | | | 4 |
| *Kiambu* | 16 | 26 | 14 | 10 | 5 | 3 | 74 |
| *Machakos* | | 5 | 2 | | | 1 | 8 |
| *Nairobi* | 10 | 25 | 22 | 13 | 1 | 8 | 79 |
| **Male** | 790 | 2143 | 1223 | 496 | 166 | 680 | 5498 |
| *Kajiado* | 28 | 112 | 66 | 28 | 8 | 3 | 245 |
| *Kiambu* | 154 | 455 | 281 | 109 | 29 | 210 | 1238 |
| *Machakos* | 104 | 259 | 137 | 58 | 21 | 234 | 813 |
| *Nairobi* | 504 | 1317 | 739 | 301 | 108 | 233 | 3202 |
| **Total** | 816 | 2202 | 1262 | 519 | 172 | 692 | 5663 |

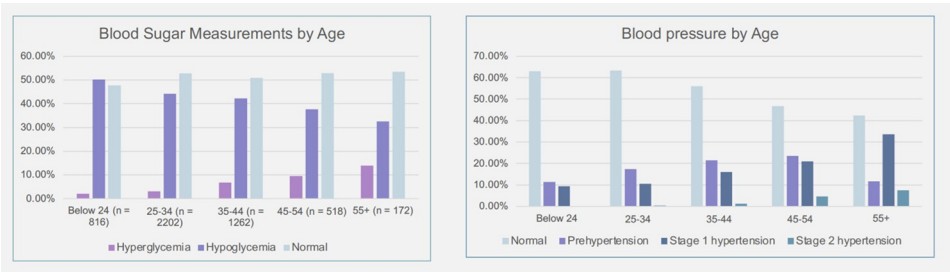

**Fig 2. Participant blood sugar and blood pressure measurements by age group.**

sugar levels (below 4 mmol/L) (Fig 2). In terms of blood pressure, 58.3% of the participants had normal blood pressure, 18.1% were prehypertensive, 14.5% were in stage 1 hypertension, and 1.3% were in stage 2 hypertension (Fig 2).

## Testing outcomes

A high proportion of the 4946 participants who underwent testing for COVID-19 reported symptoms listed on the screening checklist (Fig 3). The most common symptoms reported by participants were fatigue (62.7%), headache (33.5%) and cough of any duration (30.1%). Among the 4946 participants, 372 also reported symptoms related to both COVID-19 and TB and thus received testing for both.

## COVID-19 positivity rates

A total of 4699 samples were evaluated for the positivity rate analysis, due to missing results and one non-consent as previously explained. Positivity for SARS-CoV-2 based on the Ag-RDT was less than 0.96% overall, although positivity by county varied from 0% in Kajiado to

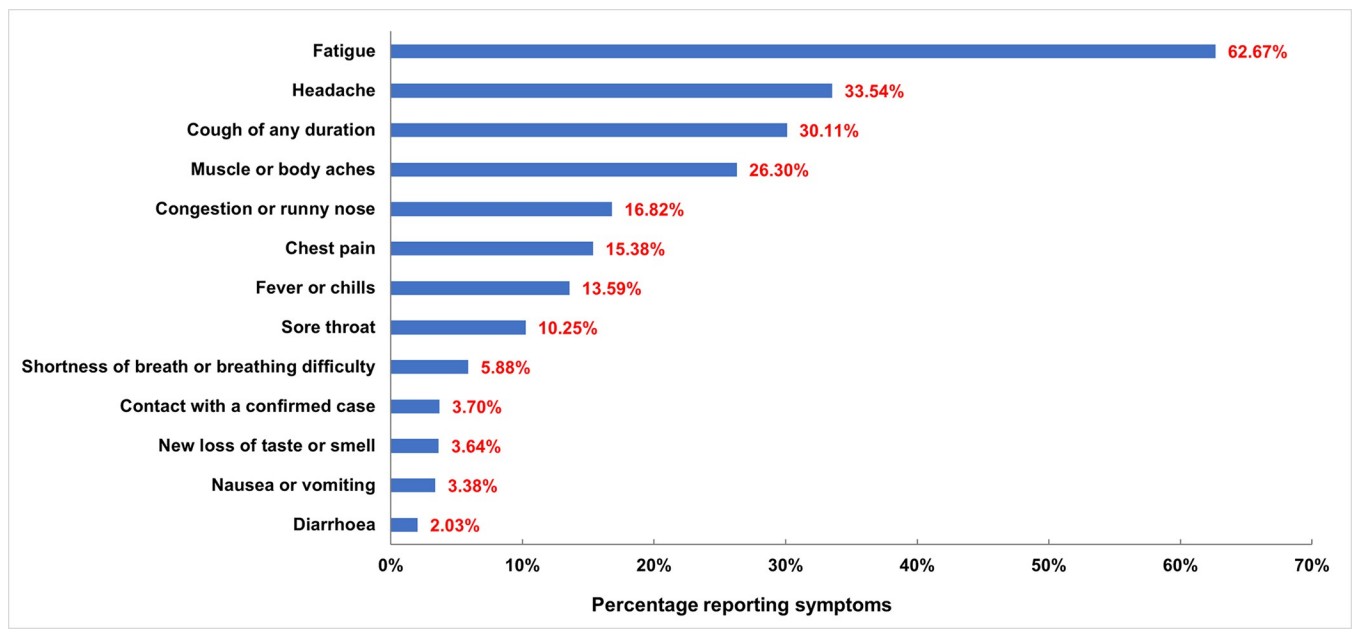

**Fig 3. Participant responses to specific screening questions.**

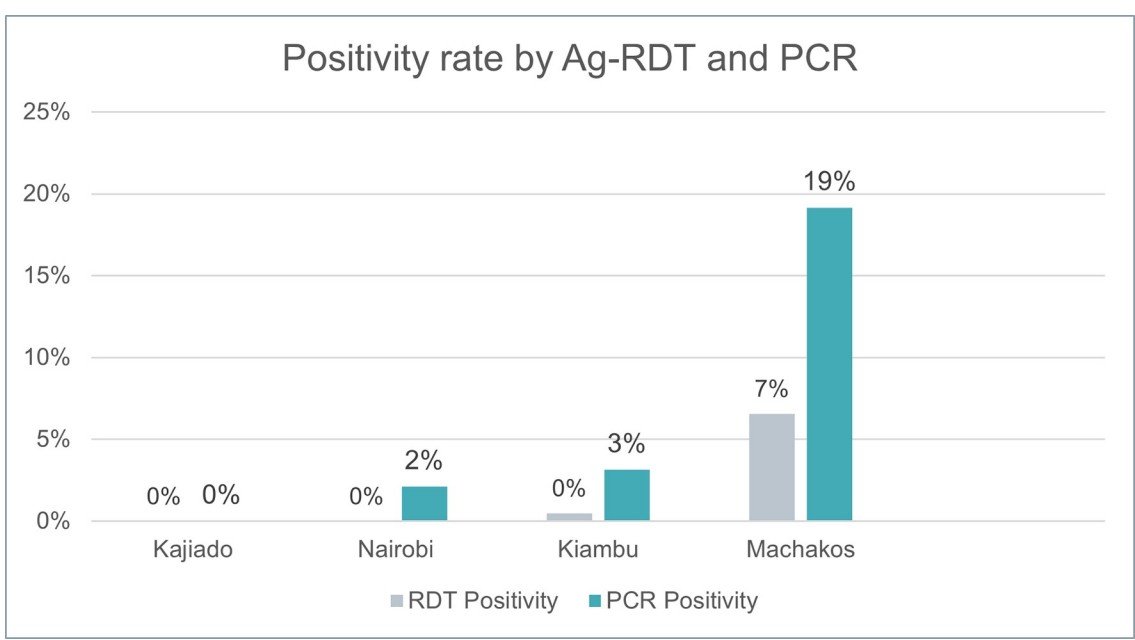

**Fig 4. Ag-RDT and RT-PCR positivity rates among participants.** Ag-RDT, rapid antigen-detecting tests; RT-PCR, reverse transcription polymerase chain reaction.

6.6% in Machakos County (Fig 4). Positivity rate for RT-PCR was 4.6% overall and varied from 0% in Kajiado County to 19.2% in Machakos County.

Per study procedures, most participants were randomly selected for PCR but a smaller proportion was selected based an automated severity prompt built into the digital tool to identify individuals considered to be at particularly high risk, based on their symptom profile. Such individuals counted for 162 out of 1021 of those who underwent testing, and had a positivity rate of 8.0% compared with 4.0% for those who were randomly selected (Table 2).

## TB positivity rates

Of the 372 sputum samples that were collected for GeneXpert analysis, 355 had valid results. Out of this, 7 samples were positive, translating to a rate of 1972 cases per 100,000 people, with a 95% confidence interval of 525 cases per 100,000 to 3418 cases per 100,000.

No cases of COVID-19 and TB co-infections were picked up during the study, either with the Ag-RDTs or PCR.

## Symptoms among those testing positive for COVID-19 or TB

The digital tool enabled end-to-end data capture that allowed for analysis of symptom distribution among COVID-19 and TB cases, with the exception of 246 records previously mentioned where human error resulted in incomplete data on testing results. Of the 45 cases who

**Table 2. PCR positivity rate by selection method.**

| Sample selection | Negative | | Positive | | Total |
|---|---|---|---|---|---|
| Random selection | 825 | 96.04% | 34 | 3.96% | 859 |
| High risk profile | 149 | 91.98% | 13 | 8.02% | 162 |
| Total | 974 | 95.40% | 47 | 4.60% | 1021 |

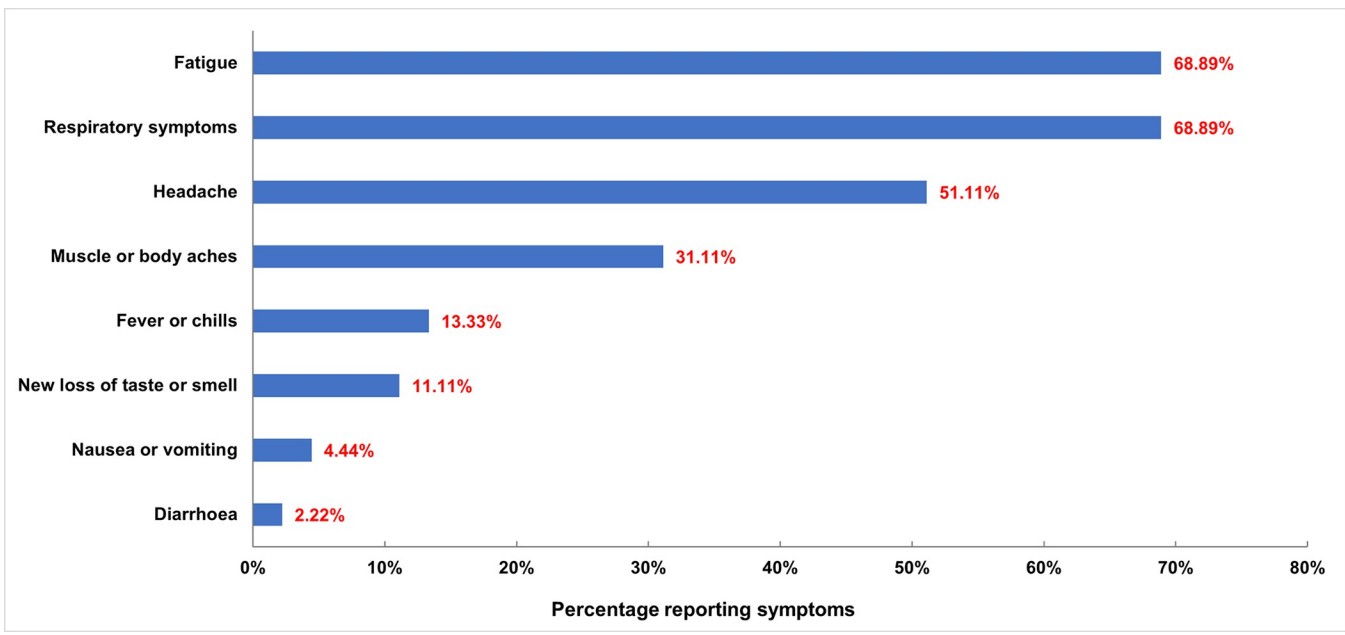

**Fig 5. Reported symptoms among participants testing positive for COVID-19 by Ag-RDT.**

tested positive for COVID-19 on the Ag-RDT, the most common symptoms were respiratory symptoms and fatigue (both at 68.9%) followed by headache (51.1%) and then muscle or body aches (31.1%). The least observed symptom was diarrhoea (2.2%), followed by nausea and vomiting (4.4%) (Fig 5). The distribution of symptoms among those testing positive for COVID-19 by RT-PCR was similar, with fatigue the most common symptom (59.6%), followed by respiratory symptoms (55.3%) headache (40.0%) and muscle and body aches (38.3%).

**Test performance.** The sensitivity of the Ag-RDT compared with RT-PCR was 27.91% (95% CI: 14.67, 41.45); specificity was 98.39% (95% CI: 97.55, 99.23). The positive predictive value of the Ag-RDT was 46.15% (95% CI: 26.99, 65.31%) and the negative predictive value was 96.51% (95% CI: 95.3, 97.72).

## Discussion

This study demonstrated that a digitally supported community-based testing model can substantially enhance COVID-19 and TB diagnosis and linkage to care among Bodaboda riders, who are at high risk of contracting COVID-19 and TB. The digital tool provided clinical decision support to guide standardized processes for selecting individuals to receive COVID-19 antigen testing, TB diagnosis and/or confirmatory COVID-19 PCR testing. With expansion of testing outside of health facilities, it can be challenging to implement traditional approaches to supervision in order to ensure adherence to clinical guidelines–research in Kenya has shown suboptimal adherence to guidelines among trained community health workers [30]. Having step-by-step clinical processes embedded in the workflow of the digital tool provides real-time guidance for health workers to encourage adherence to guidelines.

The tool also provided real-time end-to-end data capture for over 95% of participants, which supported remote monitoring and greater data quality through data validation checks built into the process. The use of digital tools also aided linkage to care, as the study was able to submit complete data on patients testing positive for COVID-19 into county health

management systems, so the MoH could follow up as needed to enroll patients in home-based care or provide further clinical services if needed. The data fields in the digital tool were therefore aligned with MoH reporting systems while integrating more detail, and all patient information captured at this community entry point was transmitted to the county TB or COVID-19 programmes through daily uploads. A limitation of the study is that it did not track the outcomes of participants after this step, except for capturing the results of people tested for TB and by PCR for COVID-19, with 100% of these results being captured.

Overall, 246 participants had incomplete data in the system due to human error (forgetting to update the Ag-RDT test result). In addition to reducing data quality, this also prevented linkage to care for patients who may have tested positive for COVID-19. This error arose because the digital tool allowed for health workers to indicate that a test was 'pending' if they needed to exit the patient record before enrolling another person while waiting for the Ag-RDT to complete. In future, inclusion of a reminder or alert if the result is not updated after 15 minutes may help to reduce incomplete records.

There was high interest in the study among Bodaboda riders, with testing sheds required to extend operating hours from 7 to 9 hours daily to accommodate higher-than-anticipated demand. Study personnel at the testing sheds also reported frequent requests for a broader range of diagnostic and treatment services that were not included in this model, and for provision of testing to spouses (which accounts for the 165 women who were enrolled in the study). Of the 231,500 Bodaboda riders registered across the Greater Nairobi Metropolitan area covered in this study, around one out of fifty (2.37%) accessed testing through this study, which offered less than ten testing points in a geographic area covering 32,000 square kilometers, and for a limited period. With expansion of testing points and time, potentially through periodic campaigns, a larger proportion of this vulnerable population could be reached.

SARS-CoV-2 positivity rates observed during the study were 1% and 4.6%, using Ag-RDTs and PCR, respectively. Positivity rates were broadly aligned with positivity rates for the counties included in the study, although there was a wide range of positivity values across the counties (range of 0–6.6% for Ag-RDTs and 0–19.2% for PCR across counties). Positivity ranges varied between counties as a result of the emerging Omicron wave in the country, which peaked in December 2021 during the study's enrollment period [31].

Enrollment of study participants in Kajiado, Kiambu and Nairobi Counties took place either before or after the peak of the Omicron wave, while enrollment in Machakos County took place just before and during Omicron wave; hence, positivity rates appear notably higher in Machakos (at 19.15%) than in other counties. This positivity rate, while high, likely underestimates the true extent of SARS-CoV-2 infection as it was based on Ag-RDT results, while most county/national estimates include a considerable contribution of results from PCR tests which are more sensitive and would therefore yield a higher positivity rate. Bodaboda hubs could therefore be an important location for rapidly increasing COVID -19 detection during surges.

The sensitivity of the Ag-RDTs in the study compared with PCR was low, at 27.9%, although specificity was high at 98.4%. Among Ag-RDT false negatives for whom PCR cycle threshold values were available, the majority were ≥30, indicating a low viral load which would negatively impact the sensitivity of the test. The positive predictive value of the Ag-RDTs (i.e. the probability that those with a positive Ag-RDT truly had COVID-19) was also modest at 46.2%, as the prevalence of COVID-19 was low across most counties (with the exception of Machakos) during the study. These findings also reflect the limitations of Ag-RDTs during low disease prevalence, which have lower sensitivity compared with RT-PCR [32, 33]. However, when considering whether to use Ag-RDTs in community testing initiatives, their lower sensitivity versus PCR should be balanced against their ease of use, lower cost, and ability to be deployed in decentralized settings.

Measures should also be taken to ensure that individuals testing negative but with high likelihood of infection, are linked to more advanced diagnostics. In this study, the PCR positivity rate among individuals flagged by the digital tool as high-risk, despite testing negative by Ag-RDTs, was 8.0% compared with 4.0% among those randomly selected. While the sample size was too small to run more detailed statistical analyses, this finding suggests that the use of digital tools to identify individuals with strong clinical predictors for COVID-19 infection can mitigate the risk of missed cases due to lower sensitivity of Ag-RDTs.

Of the 355 persons tested for TB with valid results, 7 individuals tested positive for TB using GeneXpert, which translates to a TB prevalence rate of 1972 per 100,000 people, which is over twice the national prevalence rate of 558 per 100,000 people [34]. While the sample size was also small due to fewer people meeting the criteria for TB testing than COVID-19 testing, which contributed to a wide confidence interval (525 to 3,418 case per 100,000), the lower bound of this confidence interval is still higher than the national prevalence rate, indicating higher risk of TB in this population than for the national population.

As men between the ages of 25 and 34 have the highest prevalence of TB in Kenya (809 per 100,000) and many of these comprise the Bodaboda population, targeting this population with TB case finding could also help to improve the overall case detection rate in the country. Where possible, integration of screening efforts could enable more efficient use of resources, especially given the overlap in clinical presentation with respiratory symptoms accounting for 55.3% of all who tested for COVID-19.

This study did not diagnose coinfection of TB and COVID-19 among any of the participants tested, similar to a 2020 study conducted in Nigeria on community-based TB and COVID-19 diagnosis with inclusion of digital X-ray in the screening process, which found no cases of co-infection [35]. However, a 2022 compilation by WHO of evidence from dual TB-COVID-19 screening projects reported that in the Philippines, the yield of TB cases, when testing was routinely offered to all individuals who presented for COVID-19 testing, was twice as high as the yield from standard active case finding activities, illustrating the value of integrated testing approaches [36]. The compiled evidence also showed wide variations in TB and COVID-19 infection rates across settings [36], which could be due to underlying epidemiology and/or to variations in the testing protocols used. In some settings, these protocols were not documented in detail, posing challenges to efforts to compare or synthesize evidence. The use of digital tools can help to standardize and document methodologies used in diagnostic processes, so as to allow for better interpretation of data across settings. WHO also noted that only half of the projects reviewed used digital technologies in their approach, largely to support treatment adherence or facilitate the provision of remote care [36], highlighting an opportunity to increase the use of digital solutions to support efficient models of decentralized diagnosis.

In addition to screening for COVID-19 and TB, the study also provided Bodaboda riders with a free health check, with measurement of blood pressure and blood sugar levels, and BMI. The study identified that 42% and 48% of the riders had elevated blood pressure and abnormal sugar levels, respectively. As such, the findings from this study highlight the need to increase targeted health interventions among Bodaboda riders to address the burden of non-communicable diseases experienced by this population. Given the prevalence of high blood pressure and abnormal blood sugar levels, as well as low NHIF coverage amongst Bodaboda riders (only 42.6% participants had NHIF cover), there is need for increased access to health financing through the national health insurance scheme among Bodaboda riders, which may encourage increased health-seeking and adherence to clinical guidance by this vulnerable population.

Overall, this study is important in demonstrating the feasibility and value of decentralized testing initiatives, facilitated through digital platforms. Testing remains central to controlling

both the ongoing COVID-19 pandemic and the TB epidemic in Kenya. Decentralized, community-based testing initiatives, such as the one investigated here, are particularly important for increasing access to care, particularly among high-risk and vulnerable groups who may be less able to access or afford testing otherwise. Our study shows how existing tools used as part of national disease surveillance can be adapted to support successful decentralized testing initiatives.

Other studies have also demonstrated how digital tools can be used to facilitate decentralized screening in other low- and middle-income countries. Another study in collaboration with FIND found that a digital tool combined with Ag-RDTs enhanced community-based decentralized COVID-19 testing service delivery, reporting and follow-up at taxi ranks in Johannesburg, South Africa [21]. Several digital tools, including mobile health apps, have also been developed to support with TB diagnosis [25, 37]. However, to the best of our knowledge, our study is the first to investigate the use of digital tools for screening of COVID-19 and TB, in addition to non-communicable diseases, at the community level in Kenya. It will be used to inform future national programming for bidirectional screening of COVID-19 and TB.

As discussed, a limitation of the study was that some data were missing from the final analysis, as in some cases personnel forgot to update results from participants after they were received. However, this represented a small number of total cases, with end-to-end data successfully captured for 95% of all participants. As such, these missing data are unlikely to have affected the overall conclusions. In addition, although digital tools can improve the ease of delivering decentralized testing interventions, the success of the intervention is still highly dependent on the conduct of clinical assessments and quality testing by skilled healthcare workers.

In conclusion, this study demonstrates the value of integrating screening for COVID-19 with other high-priority respiratory diseases such as TB, to improve detection of both diseases among high-risk and underserved populations. The study also shows how digital solutions can facilitate delivery of decentralized testing initiatives, which are vital to improve access to testing and control of infectious diseases like COVID-19.

## Supporting information

**S1 Checklist. STROBE statement—checklist of items that should be included in reports of observational studies.**
(DOCX)

**S1 Table. Bidirectional COVID-19 and tuberculosis screening questions.**
(DOCX)

## Acknowledgments

The authors would like to thank Kenya's Ministry of Health Director-General's Office; Public Health Emergency Operation Centre (PHEOC); Departments of Health of Nairobi, Machakos, Kiambu and Kajiado Counites; the Bodaboda Safety Association of Kenya (BAK); the JKUAT-DHARC team; and FIND, including the digital health team and the clinical trials unit which provided technical input to the study protocol. The authors would also like to thank Olukunle Akinwusi, FIND, and Medic Mobile for contributions to the customization of the digital tool. Medical writing support was provided by Talya Underwood, Principal Writer, of Anthos Communications Ltd.

## Author Contributions

**Conceptualization:** Simon Karanja, Jane Aduda, Patrick Amoth, Denver Mariga, Ian Were, Musa Mohamed, Judith Kimuyu, Samson Saigilu, Rose Wangeci, Kevin Mubadi, Joseph Ndung'u, Rigveda Kadam, Paula Akugizibwe.

**Data curation:** Simon Karanja, Jane Aduda, Henry Kissinger, Susan Mwelu.

**Formal analysis:** Simon Karanja, Jane Aduda, Khairunisa Suleiman, Paula Akugizibwe.

**Funding acquisition:** Simon Karanja.

**Investigation:** Simon Karanja, Jane Aduda, Reuben Thuo, Fred Wamunyokoli, Philip Oyier, Gideon Kikuvi, Henry Kissinger, John Gachohi, Patrick Mburugu, David Kamau, Joseph Matheri, Susan Mwelu, Joseph Machua, Denver Mariga, Ian Were, Musa Mohamed, Judith Kimuyu, Samson Saigilu, Rose Wangeci, Kevin Mubadi.

**Methodology:** Simon Karanja, Jane Aduda, Reuben Thuo, Fred Wamunyokoli, Henry Kissinger, Patrick Mburugu, David Kamau, Susan Mwelu, Patrick Amoth, Denver Mariga, Joseph Ndung'u, Paula Akugizibwe.

**Project administration:** Simon Karanja, Khairunisa Suleiman.

**Resources:** Simon Karanja, Jane Aduda, Kevin Mubadi.

**Software:** Jane Aduda, Philip Oyier, Susan Mwelu, Rigveda Kadam.

**Validation:** Simon Karanja, Jane Aduda, Reuben Thuo, Fred Wamunyokoli, Patrick Mburugu, Denver Mariga, Ian Were, Musa Mohamed, Judith Kimuyu, Samson Saigilu, Rose Wangeci, Kevin Mubadi, Paula Akugizibwe.

**Visualization:** Simon Karanja, Jane Aduda, Susan Mwelu.

**Writing – original draft:** Simon Karanja, Jane Aduda, Reuben Thuo, Fred Wamunyokoli.

**Writing – review & editing:** Simon Karanja, Jane Aduda, Reuben Thuo, Fred Wamunyokoli, Philip Oyier, Gideon Kikuvi, Henry Kissinger, John Gachohi, Patrick Mburugu, David Kamau, Joseph Matheri, Susan Mwelu, Joseph Machua, Patrick Amoth, Denver Mariga, Ian Were, Musa Mohamed, Judith Kimuyu, Samson Saigilu, Rose Wangeci, Kevin Mubadi, Joseph Ndung'u, Khairunisa Suleiman, Rigveda Kadam, Paula Akugizibwe.

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
