## [Decision Letter · Decision Letter 0]

15 Jun 2023

PONE-D-23-06769Utilization of digital tools to enhance COVID-19 and tuberculosis testing and linkage to care: a cohort study among Bodaboda motorbike riders in the Nairobi Metropolis, KenyaPLOS ONE

Dear Dr. Karanja,

Thank you for submitting your manuscript to PLOS ONE. After careful consideration, we feel that it has merit but does not fully meet PLOS ONE’s publication criteria as it currently stands. Therefore, we invite you to submit a revised version of the manuscript that addresses the points raised during the review process.

We look forward to receiving your revised manuscript.

Kind regards,

Olushayo Oluseun Olu

Academic Editor

PLOS ONE

Journal Requirements:

No competing interests to declare.

Reviewers' comments:

Reviewer's Responses to Questions

**Comments to the Author**

1. Is the manuscript technically sound, and do the data support the conclusions?

Reviewer #1: Partly

Reviewer #2: Yes

2. Has the statistical analysis been performed appropriately and rigorously? 

Reviewer #1: No

Reviewer #2: I Don't Know

3. Have the authors made all data underlying the findings in their manuscript fully available?

Reviewer #1: Yes

Reviewer #2: Yes

4. Is the manuscript presented in an intelligible fashion and written in standard English?

Reviewer #1: Yes

Reviewer #2: Yes

5. Review Comments to the Author

Reviewer #1: Technically, the design is not a "cohort study" thus the use of the term in the title could be misleading. This also applies to the description of the study design as "interventional cohort study". While a "cohort" is appropriate as a term for a study population of subgroup that was followed up, a cohort study is an analytical 'prospective' study approach to investigate risk factors of diseases or health conditions.

It is also debatable to label this an "interventional study" because, the design did not include a control group, for which a the intervention would not be applied, and the outcome variables compared. The study did not also collect baseline or pre-intervention data, which is expected of intervention study designs.

The study design may best be described as an evaluation study, using a cross-sectional design.

Regarding the study outcome variables, the authors should be more explicit in stating and describing the "various metrics" mentioned in the Analysis section (Line 320). The statement on Line 322/323 should explicitly state which measure was used as the 'gold standard' or benchmark for the other. Also, clarity should be given on how confounding was addressed at the study design stage.

The results and discussion on the performance of the Ag-RDT kits were somewhat distracting from the main purpose of the paper, including the results on the 'reported symptoms' and prevalence of Covid-19 and TB. The authors should consider limiting the paper to results of clear-cut measurements of data systems management metrics that demonstrate whether or not the digital tool/platform performed well.

The authors stated their intention to show (and in fact did conclude) that the tool enhanced linkage; other than the narrative, the analysis of quantitative data to support this should be clearly provided.

Reviewer #2: The paper is beneficial; however authors have gone to great extent to extrapolate the impact of the digital part of the work on the outcome. While the digital part is important but should be seen as just a component. After all digital components cannot predict an outcome of COVID 19 or TB rather it provides a standardized form of questionnaire, along with algorithms for recognizing suspected cases, along with possible real time data. The value of clinical assessments cannot be overemphasized, along with the skill set of HW and data manager collating this.

6. PLOS authors have the option to publish the peer review history of their article (what does this mean?). If published, this will include your full peer review and any attached files.

Reviewer #1: **Yes: **Seye Babatunde

Reviewer #2: **Yes: **Sylvester Maleghemi

---

## [Author Response · Author response to Decision Letter 0]

28 Jul 2023

Comments from Academic Editor

Response: Thank you, we have ensured that our manuscript meets the journal’s style requirements. 

No competing interests to declare.

Response: Thank you. We have updated our competing interests statement accordingly and added this to the cover letter (copied here for reference):

J.N., K.S., R.K., and P.A. declare that they are employed by FIND. The other authors have declared that no competing interests exist. 

Response: Thank you, Professor Karanja has validated his ORCID ID in Editorial Manager. 

Review Comments to the Author

Reviewer #1: Technically, the design is not a "cohort study" thus the use of the term in the title could be misleading. This also applies to the description of the study design as "interventional cohort study". While a "cohort" is appropriate as a term for a study population of subgroup that was followed up, a cohort study is an analytical 'prospective' study approach to investigate risk factors of diseases or health conditions.

It is also debatable to label this an "interventional study" because, the design did not include a control group, for which a the intervention would not be applied, and the outcome variables compared. The study did not also collect baseline or pre-intervention data, which is expected of intervention study designs.

The study design may best be described as an evaluation study, using a cross-sectional design.

Response: Thank you, we have updated the title of the manuscript and description of the study in the methods section to reflect the comments above. The study is now described as a cross-sectional evaluation study. 

Regarding the study outcome variables, the authors should be more explicit in stating and describing the "various metrics" mentioned in the Analysis section (Line 320). The statement on Line 322/323 should explicitly state which measure was used as the 'gold standard' or benchmark for the other. Also, clarity should be given on how confounding was addressed at the study design stage.

Response: Thank you, we have updated line 322 to clarify which metrics were recorded and explicitly state that PCR was the gold standard for evaluation of Ag-RDT performance. This section now reads:

“Various metrics were computed on the number of participants at each stage of the study, including the number of participants who visited the sites and underwent screening for COVID-19 and TB, and the proportion of participants with complete records captured in the digital tool. Also recorded was the medical background of participants such as blood sugar and blood pressure levels. Results from the COVID-19 and TB tests were recorded in terms of the number testing positive, negative and inconclusive. COVID-19 positivity rates were recorded for Ag-RDT and RT-PCR testing methods, while TB positivity rates were recorded based on GeneXpert analysis of sputum samples. Ag-RDT test performance was calculated in terms of sensitivity, specificity, positive predictive value and negative predictive value, evaluated against the “gold standard” RT-PCR. Symptoms were also recorded among those testing positive for COVID-19 and TB.”

We have also expanded on the statement about confounding to provide further details – updated paragraph copied below for reference: 

“Confounding was addressed mainly at the study design stage. In particular, as the study focused on a specific group (male Bodaboda riders mainly between 25 to 45 years of age in the Nairobi Metropolis), sex, occupation, age and geographical location were largely controlled. Nasal sample collection was conducted by laboratory officers/technologists who had received training on sample collection from trainers (government laboratory officers who had previously been trained on COVID-19 testing), in line with national guidelines. PCR confirmatory tests were carried out in centralized, national accredited government diagnostic laboratories in designated level 4 and 5 health facilities, which ensured standardized testing. Further, data collection was carried out by personnel who were trained using standard protocols and guidelines, supervised by trainers.” 

The results and discussion on the performance of the Ag-RDT kits were somewhat distracting from the main purpose of the paper, including the results on the 'reported symptoms' and prevalence of Covid-19 and TB. The authors should consider limiting the paper to results of clear-cut measurements of data systems management metrics that demonstrate whether or not the digital tool/platform performed well.

The authors stated their intention to show (and in fact did conclude) that the tool enhanced linkage; other than the narrative, the analysis of quantitative data to support this should be clearly provided.

Response: Thank you, while we agree that the focus of the paper is on the use of digital tools, we believe it is useful to include the Ag-RDT performance data to add to the literature on Ag-RDT performance in diverse real-world settings. We believe the information on sensitivity, specificity and positive/negative predictive value may be useful to inform decisions on the use of Ag-RDTs in similar interventions. 

In addition, we have revised the statement around the value of digital tools supporting linkage to care on lines 448–451 to say “The use of digital tools also aided in linking patients to care, as the study was able to submit complete data on patients testing positive for COVID-19 into county health management systems, so the MoH could follow up as needed to enroll patients in home-based care or provide further clinical services if needed”. This reflects the findings from the study and experiences of personnel involved in the study. 

Reviewer #2: The paper is beneficial; however, authors have gone to great extent to extrapolate the impact of the digital part of the work on the outcome. While the digital part is important but should be seen as just a component. After all digital components cannot predict an outcome of COVID 19 or TB rather it provides a standardized form of questionnaire, along with algorithms for recognizing suspected cases, along with possible real time data. The value of clinical assessments cannot be overemphasized, along with the skill set of HW and data manager collating this.

Response: Thank you, we agree with your point that the digital tool is a component of a successful screening intervention. As such, our main conclusion (as noted in the abstract) is that integrated diagnosis [of COVID-19 and TB] can be delivered effectively in communities, with the support of digital tools, to maximize access. In the paragraph on study limitations, we have added a sentence to note the value of clinical assessments as per your comment (lines 569 to 568). In the conclusion, we have also amended the final sentence to note that digital solutions can facilitate delivery of decentralized testing initiatives.

---

## [Editor Report · Decision Letter 1]

11 Aug 2023

Utilization of digital tools to enhance COVID-19 and tuberculosis testing and linkage to care: a cross-sectional evaluation study among Bodaboda motorbike riders in the Nairobi Metropolis, Kenya

PONE-D-23-06769R1

Dear Dr. Karanja,

We’re pleased to inform you that your manuscript has been judged scientifically suitable for publication and will be formally accepted for publication once it meets all outstanding technical requirements.

Kind regards,

Olushayo Oluseun Olu

Academic Editor

PLOS ONE
---

## [Editor Report · Acceptance letter]

31 Aug 2023

PONE-D-23-06769R1 

Utilization of digital tools to enhance COVID-19 and tuberculosis testing and linkage to care: a cross-sectional evaluation study among *Bodaboda* motorbike riders in the Nairobi Metropolis, Kenya 

Dear Dr. Karanja:

I'm pleased to inform you that your manuscript has been deemed suitable for publication in PLOS ONE. Congratulations! Your manuscript is now with our production department. 

Kind regards, 

on behalf of

Dr. Olushayo Oluseun Olu 

Academic Editor

PLOS ONE